**DOI: 10.1038/ncomms14815**　　**OPEN**

# Controllable conversion of quasi-freestanding polymer chains to graphene nanoribbons

Chuanxu Ma[1,*], Zhongcan Xiao[2,*], Honghai Zhang[1], Liangbo Liang[1], Jingsong Huang[1,3], Wenchang Lu[1,2], Bobby G. Sumpter[1,3], Kunlun Hong[1], J. Bernholc[1,2] & An-Ping Li[1]

In the bottom-up synthesis of graphene nanoribbons (GNRs) from self-assembled linear polymer intermediates, surface-assisted cyclodehydrogenations usually take place on catalytic metal surfaces. Here we demonstrate the formation of GNRs from quasi-freestanding polymers assisted by hole injections from a scanning tunnelling microscope (STM) tip. While catalytic cyclodehydrogenations typically occur in a domino-like conversion process during the thermal annealing, the hole-injection-assisted reactions happen at selective molecular sites controlled by the STM tip. The charge injections lower the cyclodehydrogenation barrier in the catalyst-free formation of graphitic lattices, and the orbital symmetry conservation rules favour hole rather than electron injections for the GNR formation. The created polymer–GNR intraribbon heterostructures have a type-I energy level alignment and strongly localized interfacial states. This finding points to a new route towards controllable synthesis of freestanding graphitic layers, facilitating the design of on-surface reactions for GNR-based structures.

[1] Center for Nanophase Materials Sciences, Oak Ridge National Laboratory, Oak Ridge, Tennessee 37831, USA. [2] Department of Physics, North Carolina State University, Raleigh, North Carolina 27695, USA. [3] Computer Science and Mathematics Division, Oak Ridge National Laboratory, Oak Ridge, Tennessee 37831, USA. * These authors contributed equally to this work. Correspondence and requests for materials should be addressed to A.-P.L. (email: apli@ornl.gov).

In the pursuit of atomically precise and bottom-up fabrication of graphene-based electronics[1–3], graphene nanoribbons (GNRs) with a variety of widths[4,5], edge structures[6,7] and heterojunctions[8,9] have been synthesized with self-assembled molecular precursors on different catalytic metal substrates, such as Au (refs 10–12), Ag (ref. 13) and Cu (refs 14,15). Surface-assisted cyclodehydrogenations[16–18], a key step in GNR formation, appear to take place only on the catalytic metal substrates[19,20]. Efforts of growing GNRs from molecular precursors directly on an insulating $TiO_2$ substrate only succeeded in deriving polymerization but not the cyclodehydrogenation[20]. The metallic substrates are found to be essential not only for the GNR growth process, but also for the electronic behaviours of synesized GNRs. The orbital hybridization between substrate and edge atoms can largely affect the predicted edge states and magnetism of GNRs[21,22], and the dielectric screening interaction from the substrate can greatly modify the quasiparticle bandgaps of GNRs. For example, the armchair GNRs with a width of seven carbon (7-aGNRs) on Au(111) are reported to have an energy gap ranging from 1.8 to 2.8 eV (refs 4,10,13,23–26). These values are much smaller than expected from electronic structure calculations within many-body perturbation theory in the GW approximation, which predicts a gap of ~3.7 eV (ref. 27). Controlling the substrate interaction is thus considered a prerequisite for studying the on-surface synthesis process and accessing the intrinsic electronic structure of GNRs.

Here we focus on the controllable conversion of quasi-freestanding polymers to form atomically precise armchair graphene nanoribbons, 7-aGNRs. To decouple their electronic structure from the metal substrate, we grow the polymers atop first-layer (1st-layer) GNRs that are in direct contact with the metal substrate. The polymers are isolated from the metal substrate by the 1st-layer GNRs, leading to their quasi-freestanding nature. Using scanning tunnelling microscopy (STM), we find that electronic decoupling of the polymer can greatly slow down the cyclodehydrogenation and an STM tip can be used to inject charges at the selected molecular sites to trigger the reaction and thus create intraribbon heterojunctions. Based on nudged elastic band (NEB) simulations, we reveal a hole-assisted cyclodehydrogenation reaction path that points to an avenue towards the controllable on-surface synthesis of freestanding GNRs and precise intraribbon heterojunctions.

## Results

### Synthesis of quasi-freestanding polymer chains.
The polyanthrylene chains were synthesized on Au(111) using 10,10′-dibromo-9,9′-bianthryl (DBBA) molecules as precursors with a bottom-up method described by Cai et al.[11] (Methods). Figure 1a schematically illustrates the stepwise annealing process for growing the 7-aGNRs and quasi-freestanding polymer chains atop the GNRs. With a molecule coverage $\theta > 1$, the sample is subsequently annealed to enable colligation/polymerization (at 470 K) (Supplementary Fig. 1) and cyclodehydrogenation/graphitization (at 670 K), resulting in the polymer chains atop the GNRs (Fig. 1b). The STM images in Fig. 1c,d show the 1st-layer 7-aGNRs adsorbed on Au(111) and the second-layer (2nd-layer) polymer chains, respectively. The 2nd-layer polymer chains mainly grow along the GNRs, showing a period of about 8.4 Å, consistent with the simulated STM image shown in Fig. 1e and with previous reports for polymers directly adsorbed on Au(111) surface[11]. The sub-hexagon-ring features in the STM image (Fig. 1f) are reproduced by the simulated charge density distribution of the highest occupied crystal orbital of the polymer (HOCO$_p$, Fig. 1g). The 2nd-layer polymer chains are

effectively decoupled from the Au substrate due to the GNRs underneath, which enables imaging of the quasi-atomic polymer structures.

The freestanding nature of the 2nd-layer polymer chains becomes clearer by comparing their geometric and electronic structures with the 1st-layer. The STM image in Fig. 2a shows a 2nd-layer polymer chain and a 2nd-layer GNR on two adjacent Au(111) terraces, with height profiles shown in Supplementary Fig. 2. The apparent height of the 2nd-layer polymer ($\sim 4.3$ Å) is greater than that of the 1st-layer polymer directly adsorbed on Au ($\sim 3.9$ Å, Supplementary Fig. 3), and so is the 2nd-layer GNR ($\sim 2.9$ Å) as compared to the 1st-layer GNR ($\sim 2.1$ Å). Figure 2b shows the tunnelling conductance d$I$/d$V$ spectra acquired at different locations marked in Fig. 2a. The 2nd-layer polymer (location 1) exhibits a large energy gap of about 4.3 eV with the highest occupied and lowest unoccupied crystal orbitals of the polymer (HOCO$_p$ and LUCO$_p$) in the density of states (DOS) at sample voltage $V_s = -2.1$ and $+2.2$ V, respectively. This gap is significantly greater than that for the 1st-layer polymer with a bandgap about 3.4 eV (Supplementary Fig. 3), indicating reduced dielectric screening of the substrate[10,24,28]. Note that the image in Fig. 1e acquired at $V_s = -2$ V is very close to the HOCO$_p$ and thus it reflects the intrinsic electronic structures of the polymer chain. For the 1st-layer GNR (location 4), the HOCO$_g$ and LUCO$_g$ are located at $V_s = -0.9$ and $+1.4$ V, respectively, with a bandgap about 2.3 eV, consistent with previous reports[24,25,28]. The 2nd-layer GNR (location 2) shows a larger gap of about 2.6 eV. It is found that the gap in the 2nd-layer GNR is generally about 0.1–0.4 eV greater than that in the 1st-layer GNR, where the gap difference between the two layers is comparable with previously reported difference ($\sim 0.5$ eV) between GNR on an insulator NaCl (with a bandgap $\sim 2.8$ eV)[29] and on Au (with a bandgap $\sim 2.3$ eV)[24]. Moreover, the d$I$/d$V$ spectra from both the 2nd-layer polymer and the 2nd-layer GNR show cleaner gaps with lower densities of in-gap states than the 1st-layer GNR. Thus the 1st-layer GNR, similar to graphene[30], can largely isolate the 2nd layers from the Au(111) substrate and render the 2nd-layer polymers quasi-freestanding.

### Thermally induced domino-like polymer to GNR conversion.
The existence of the 1st-layer GNRs significantly suppresses the catalytic effect of Au substrate and slows down the cyclodehydrogenation reactions in the 2nd-layer polymers, which facilitates the control and evaluation of the cyclodehydrogenation process. After annealing at 670 K, polymers only exist on the 2nd layer atop the GNRs, while the 1st-layer polymers have all been converted into GNRs. The full conversion of the 2nd-layer GNRs can occur when they have direct local contacts with the Au substrate, such as location 3 in Fig. 2a. The d$I$/d$V$ curve measured at location 3 (Fig. 2b) is similar to that of the 1st-layer GNRs (location 4). Without the direct Au contact, only partially converted 2nd-layer polymers are observed with GNR tails (Fig. 2c, more examples in Supplementary Fig. 4), which may be attributed to a charge transfer effect promoted by work function mismatch between the polymer, GNR and the Au substrate (as explained in the Supplementary Fig. 3). The GNR tail has the characteristic height ($\sim 2.9$ Å) and tunnelling spectra of the 2nd-layer GNR (Supplementary Fig. 5). Moreover, the GNR tail shows enhanced DOS at the edges compared to the 1st-layer GNR, similarly to the GNR on an insulating substrate[29]. The GNR segment always appears at an end of the polymer chain, indicating that the cyclodehydrogenation prefers to start at the polymer end and then propagate along the polymer chain. Such a domino-like cyclodehydrogenation process can drastically lower the reaction energy barrier[31] during thermal annealing as illustrated in Fig. 2d.

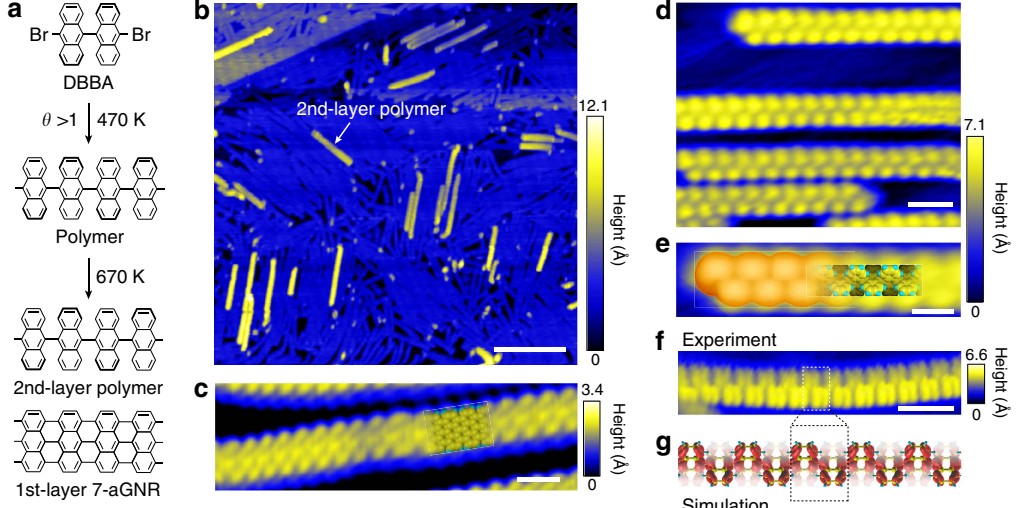

**Figure 1 | Bottom-up synthesis of polymer chains on armchair graphene nanoribbons with a width of seven carbon (7-aGNRs).** (**a**) Sketch for synthesis of the second-layer (2nd-layer) polyanthrylene chains on 7-aGNRs from 10,10′-dibromo-9,9′-bianthryl (DBBA) molecules with stepwise annealing at 470 and 670 K, respectively. (**b**) Large-area scanning tunnelling microscopy (STM) image showing the polymer chains on 7-aGNRs (sample voltage $V_s = -2$ V, tunnelling current $I_t = 60$ pA). Scale bar, 20 nm. (**c**) High-resolution STM image of the first-layer (1st-layer) 7-aGNR ($V_s = -0.6$ V, $I_t = 100$ pA) superposed with an atomic structure. Scale bar, 1 nm. (**d**) Small-scale STM image of the 2nd-layer polymer chains ($V_s = +1$ V, $I_t = 60$ pA). Scale bar, 2 nm. (**e**) The simulated STM image and an atomic structure of the polymer superposed on the magnified image of the top polymer chain in **d**. Scale bar, 1 nm. (**f**) High-resolution STM image showing the detailed structure of the polymer ($V_s = -2$ V, $I_t = 10$ pA). Scale bar, 2 nm. (**g**) Charge density distribution of the highest occupied crystal orbital of the polymer (HOCO$_p$). Dashed boxes mark the polymer unit in the polymer.

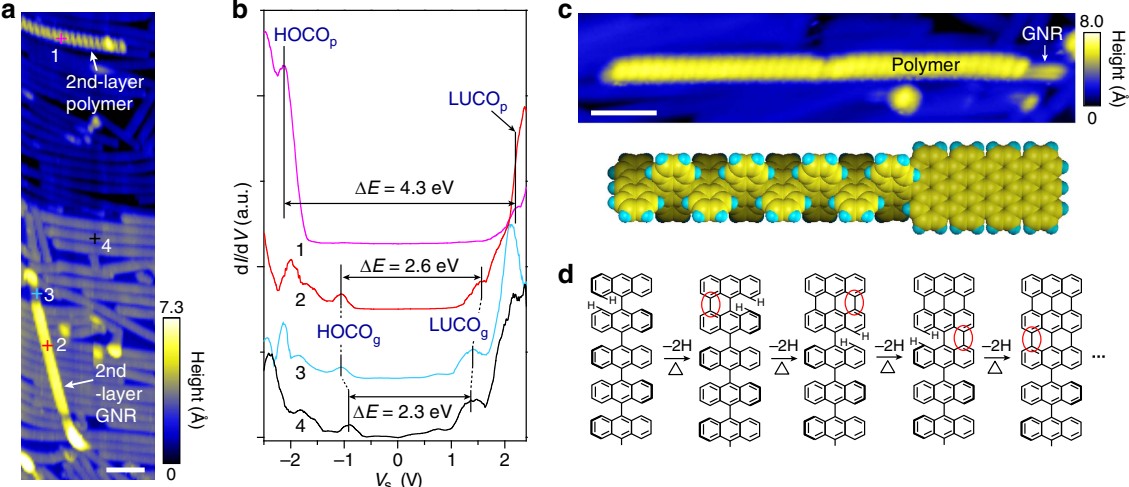

**Figure 2 | Domino-like thermally induced cyclodehydrogenation.** (**a**) STM image showing a 2nd-layer polymer and 7-aGNR, marked with white arrows ($V_s = -2$ V, $I_t = 100$ pA). Scale bar, 5 nm. (**b**) Representative differential conductance, d$I$/d$V$, curves acquired on the cross marked sites 1–4 in **a**, respectively ($V_s = -2$ V, $I_t = 100$ pA). The highest occupied and lowest unoccupied crystal orbitals of the polymer (HOCO$_p$ and LUCO$_p$) are approximately −2.1 and +2.2 eV, respectively. The HOCO$_g$ is approximately −1.0 eV and LUCO$_g$ is approximately +1.6 eV for the 2nd-layer GNR, while respectively approximately −0.9 and +1.4 eV for the 1st-layer GNR. (**c**) STM image showing an intraribbon heterojunction of a polymer chain with a GNR tail ($V_s = -2$ V, $I_t = 100$ pA), as illustrated by the schematic. Scale bar, 5 nm. (**d**) Sketch of the domino-like cyclodehydrogenation during thermal annealing. Hydrogen atoms in each step are highlighted.

This observation is in contrast with the previously reported one-side-domino conversions for polymers directly adsorbed on Au(111) (ref. 32; Supplementary Fig. 6).

**STM tip-induced polymer to GNR conversion.** To facilitate the cyclodehydrogenation reaction in the freestanding polymer chains, an STM tip is used to inject charge carriers at selected molecular sites. Figure 3a shows a 2nd-layer polymer chain on 7-aGNRs, on which a series of d$I$/d$V$ spectra are acquired along the polymer chain by moving the STM tip step-by-step

(5 Å intervals) beyond the top end of the polymer. The d$I$/d$V$ spectra are displayed in Fig. 3b, where curves 1–7 are on the polymer chain and 8–10 on the 1st-layer GNR. On the polymer chain, while curves 1–3 exhibit typical electronic features of the 2nd-layer polymer with LUCO$_p$ at $V_s = +2.1$ V (black dashed line), a new peak at $V_s = +1.7$ V emerges in curves 4–7, corresponding to the LUCO$_g$ of the GNR in curves 8–10 (marked with red dashed line). Thus at locations 4–7 near the end of the polymer chain, the polymer has been converted into GNR during the d$I$/d$V$ measurements. Indeed, a GNR tail becomes discernable

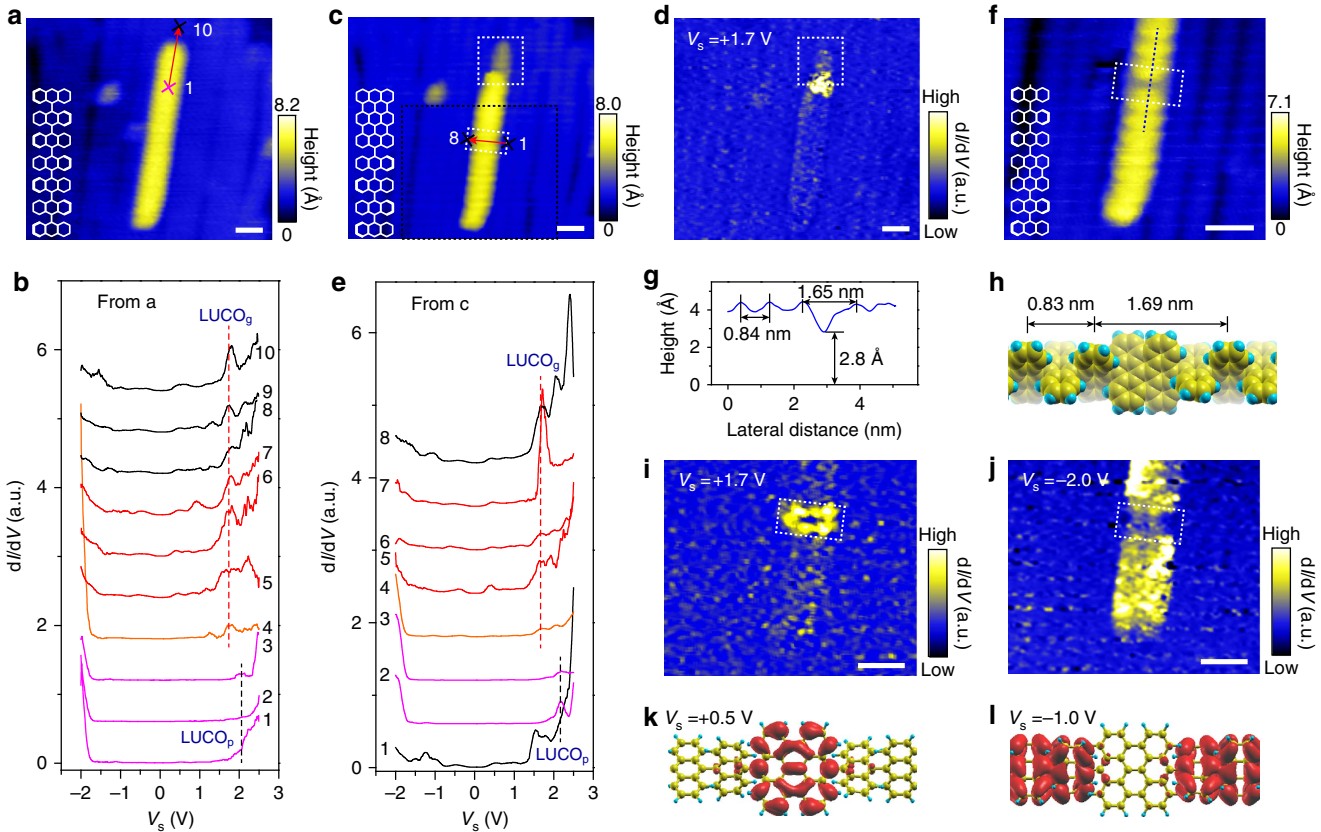

**Figure 3 | Formation of GNR segments in polymer chain induced by tunnelling electrons.** (**a**) STM image of a 2nd-layer polymer chain ($V_s = -2$ V, $I_t = 60$ pA). (**b**) d$I$/d$V$ curves sequentially acquired along the red-arrow line in **a** from equally separated site 1 to 10 ($V_s = -2$ V, $I_t = 60$ pA). Sites 1–7 are on the polymer chain. Sites 8–10 are on the 1st-layer GNR. The dashed black line marks LUCO$_p$ of the polymer in curves 1–3. The dashed red line marks the peak in curves 4–7, showing same position as LUCO$_g$ of the GNR in curves 8–10. (**c**) STM image showing a GNR segment formed at the top end of the polymer chain (white box) ($V_s = +1.7$ V, $I_t = 60$ pA). (**d**) d$I$/d$V$ mapping at $V_s = +1.7$ V ($I_t = 60$ pA), within the same area as **c**. (**e**) d$I$/d$V$ curves sequentially acquired along the red-arrow line in **c** from equally separated sites 1–8 ($V_s = -2$ V, $I_t = 60$ pA). Sites 1 and 8 are on the 7-aGNRs. Sites 2–7 are on the polymer chain. The dashed black line marks LUCO$_p$ of polymer in curves 2 and 3. The dashed red line marks the peak in curves 4–7, showing same position as LUCO$_g$ of GNRs in curves 1 and 8. (**f**) STM image of the black box marked region in **c** with a defect (white box) ($V_s = +1.7$ V, $I_t = 60$ pA). Insets in **a,c,f**: schematics of the polymer chain before and after manipulations. (**g**) Profile along the dashed line in **f**. (**h**) Atomic structure of a polymer chain embedded with a short GNR segment. (**i,j**) d$I$/d$V$ mapping at $V_s = +1.7$ and $V_s = -2$ V respectively ($I_t = 60$ pA), within the same area as **f**. (**k,l**) Charge density distribution of the states in the intraribbon heterojunction at $+0.5$ eV (**k**) and $-1$ eV (**l**), respectively. All scale bars, 2 nm.

after the d$I$/d$V$ measurements as shown in Fig. 3c. The local conversion of the polymer creates a polymer/GNR junction, and the d$I$/d$V$ mapping at $V_s = +1.7$ V shows strong localized interfacial states at the junction (Fig. 3d).

The charge injection effect on the cyclodehydrogenation process is corroborated by another set of experiments where d$I$/d$V$ curves are sequentially acquired along the red arrow across the polymer chain in Fig. 3c. Here locations 1 and 8 are on GNRs and 2–7 on the polymer. As shown in Fig. 3e, while d$I$/d$V$ curves 2 and 3 exhibit the typical electronic features of the polymer, a new peak (red dashed line) corresponding to the LUCO$_g$ of the GNR emerges in curves 4–7. The newly formed GNR segment appears like a defect in polymer chain after the d$I$/d$V$ measurements (Fig. 3f). The defect has a height of about 2.8 Å (Fig. 3g) that is very close to that of the 2nd-layer 7-aGNR (2.9 Å), and a width is about 1.65 nm that is about twice the period in the polymer (8.4 Å). Thus, the STM tip treatment has converted one polyanthrylene unit into GNR segment, which consists of three hexagon rows of 7-aGNR with a proposed structural model shown in Fig. 3h. The measured electronic states at $+1.7$ eV (Fig. 3i) are strongly localized at the interfaces between the GNR and the polymer, while the states at $-2$ eV (Fig. 3j) are suppressed at the junction as compared to those in

the polymer. According to density functional theory calculations, the states above the Fermi level (for example, at $+0.5$ eV, Fig. 3k) are mainly located in the GNR segment, while those below the Fermi level (for example, at $-1$ eV, Fig. 3l) are in the polymer segment. Notably, the energy differences between the experiments and the calculations may arise from an underestimate of the bandgap in density functional theory calculations[9,33] (Supplementary Fig. 7). Since the LUCO (HOCO) in the GNR segment is lower (higher) than that in the polymer segment, the polymer–GNR heterojunction is analogous to a type-I semiconductor junctions with a band misalignment of $\sim 0.5$–$0.8$ eV.

The tip-induced cyclodehydrogenation is examined by comparing the effects of electron and hole injections from an STM tip. The tip-treatment process is illustrated in Fig. 4a. At a selected site, the STM feedback loop is turned off (tip treatment with the STM feedback loop on can also work, Supplementary Fig. 9) and then a current pulse is applied between the tip and the sample. With a negative sample bias in the range of $V_s = -2$ to $-4$ V, hole injections are found to induce cyclodehydrogenation in the freestanding polymer chains. However, electron injections with a sample bias in the range of $V_s = +2.5$ to $+6$ V just damage the polymers without triggering cyclodehydrogenation

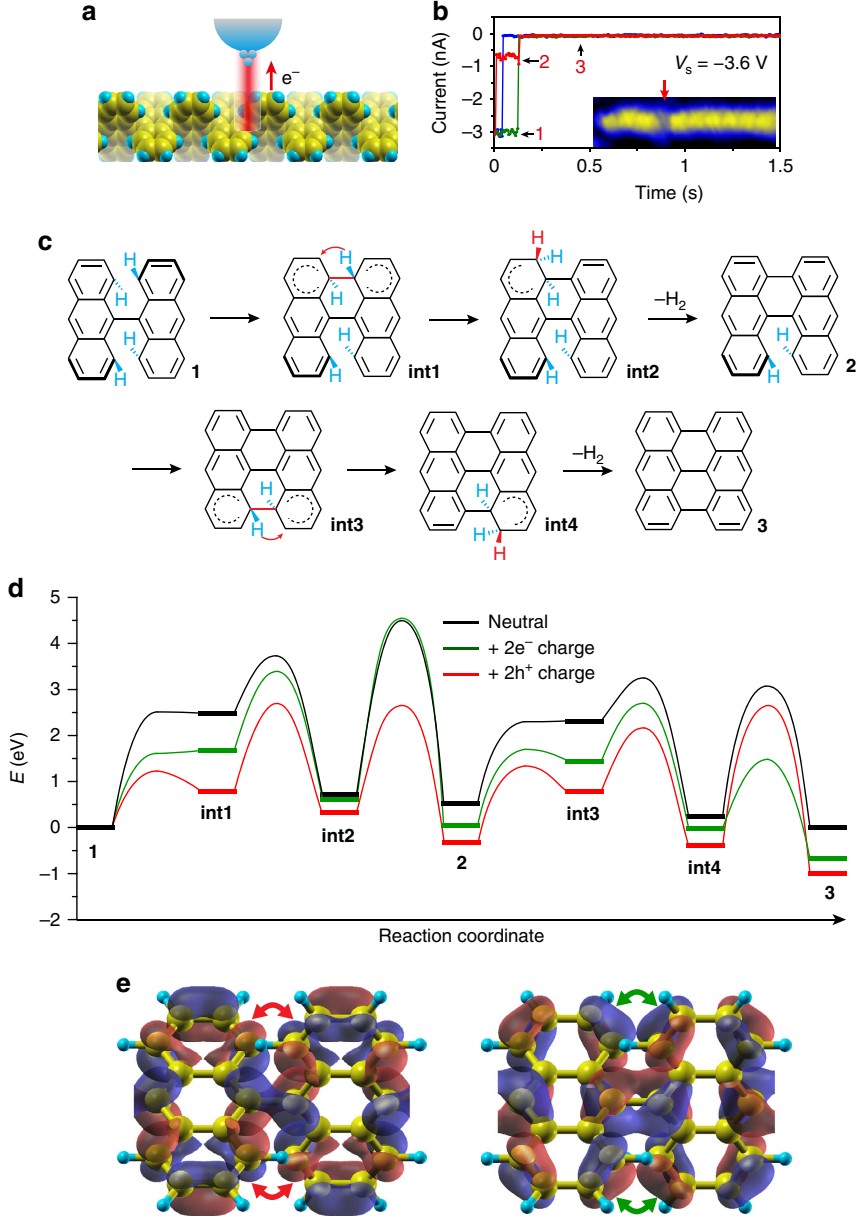

**Figure 4 | Mechanism of the holes-assisted cyclohydrogenation induced by an STM tip.** (**a**) Sketch of applying a hole pulse to a polymer chain. (**b**) Three typical tunnelling current–time ($I_t$–$t$) curves during pulses ($V_s = -3.6$ V, $t = 1.5$ s) with feedback loop off ($V_s = -2$ V, $I_t = 100$ pA). The three terraces are marked as 1–3. Inset: STM image of forming a GNR segment (red arrow) in a polymer chain ($V_s = -2$ V, $I_t = 100$ pA). (**c**) Proposed cyclohydrogenation reaction path, with 1 as the initial, 2 after one-side cyclohydrogenation and 3 as final, while int1–int4 are intermediates. A visualization of the entire reaction path including transition states can be found in Supplementary Fig. 10. (**d**) Energy diagrams of cyclohydrogenation in vacuum for neutral (black), two-electron (green) and two-hole (red) assisted bianthrylenes, respectively. (**e**) Simulated charge density distributions of $LUCO_p$ and $HOCO_p$ in the polymer where the blue and red colours indicate the different signs of wavefunctions. The out-of-phase overlap (that is, opposite signs) in $LUCO_p$ is marked by red arrows and the in-phase overlap (that is, same signs) in $HOCO_p$ is marked by green arrows, indicating that the C–C bond formation in the cyclohydrogenation is symmetry forbidden in $LUCO_p$ but symmetry allowed in $HOCO_p$.

(see Supplementary Note 1 on the yield of tip treatments with different operational parameters). The measured tunnelling current ($I_t$) is shown in Fig. 4b as a function of time ($t$) for three different tip treatment processes with a sample bias $V_s = -3.6$ V. Obvious drops of the current from terrace 1 to terrace 3 with a non-zero smaller value are observed in all three curves, indicating the occurrence of the cyclohydrogenation event. An additional terrace, terrace 2, is seen in one of the curves (red), suggesting an additional state in the cyclohydrogenation process. As the polymer is slightly taller than the GNR, the conversion of the polymer to GNR enlarges the tip–sample

distance and thus leads to a current drop. After the pulse treatment, local conversion of the polymer to GNR can be seen in the STM image shown in the inset of Fig. 4b (more examples in Supplementary Figs 8 and 9). During the experiment, polymer segments with up to three bianthrylene units (~2.5 nm) can be fully converted to GNRs by a single pulse.

**Hole-assisted cyclohydrogenation mechanism.** Figure 4c shows the proposed three-state reaction path for the cyclohydrogenations rationalized by NEB simulations[34], in correspondence with the observed three terraces in the $I_t$–$t$ curves

(Fig. 4b). Besides the three main reaction states, multiple transition states (TS) and intermediate states (int) are identified based on the NEB simulations in vacuum (Supplementary Fig. 10). In the initial state 1, the neighbouring anthrylene units first rotate around the single C–C bond, allowing two benzyne groups ($C_6H_4$) on the same side to form a single C–C bond giving int1. This step is followed by a [1,3]-sigmatropic H migration to an edge C atom giving int2. Subsequently, the elimination of a $H_2$ molecule leads to the rearomatization of the system giving state 2. Likewise, the benzyne groups on the other side repeat the process to form a graphitic lattice in state 3. Figure 4d shows the corresponding reaction energy diagrams for the neutral and two-electron- and two-hole-assisted bianthrylenes. Compared to the neutral case, the total barrier from state 1 to state 2 can be reduced from 4.5 to 2.8 eV by injecting two holes, whereas the barrier remains essentially the same for the electron injection case. In the key step of the C–C bond formation (state 1 to int1), hole injections reduce the barrier from 2.5 to 1.2 eV. Although electron injections can also reduce this barrier, the corresponding int1 would not be stable because the transition from int1 back to state 1 has a zero-energy barrier, similarly to the neutral case. Thus, hole injections can significantly facilitate cyclodehydrogenations as compared to neutral and electron injection processes, as observed in the experiment. For the subsequent reaction from state 2 to state 3, the unphysical neutral and electron injection processes are excluded from the discussion. For the hole injection case, the highest barrier is comparable to that from state 1 to state 2. However, state 3 is stabilized with respect to state 2 by about 0.7 eV, even more than that between state 2 and state 1 ($\sim 0.3$ eV). These results suggest that state 3 can be thermodynamically favoured over state 2, implying that once state 2 is formed, it may be converted to state 3 with ease, showing a cooperative cyclodehydrogenation. Indeed, state 2 corresponding to the terrace 2 in the $I_t$–$t$ curve is rarely detected when state 3 is observed during the tip treatment (Fig. 4b). We also calculated the single-charge injection cases and found that the two-hole injection mechanism is more favourable (Supplementary Fig. 11).

The hole-assisted cyclodehydrogenations are believed to be associated with inelastic tunnelling at the polymer $HOCO_P$ resonance state[35]. Figure 4e shows the simulated charge density distribution of $LUCO_P$ and $HOCO_P$ in a polymer, where the different signs as represented by blue and red colours are adopted from the orbital wavefunctions. According to the Woodward–Hoffmann rules for orbital symmetry conservation in pericyclic reactions[36,37], the formation of a C–C bond through electron injections into the $LUCO_P$ state is symmetry forbidden (red arrows) due to the opposite phase relationship of wavefunctions, while it is symmetry allowed (green arrows) through hole injections into the $HOCO_P$ state as the involved wavefunctions have the same phase relationship. Such a difference in orbital symmetries may be responsible for the different reaction barriers shown in Fig. 4d, especially between state 1 and int1. Interestingly, the hole-assisted cyclodehydrogenations are similar to the well-known Scholl reaction[38] (Supplementary Fig. 12). In organic chemistry, oxidants such as $FeCl_3$ are often used to extract electrons (inject holes) in the Scholl reaction[39,40], with which GNRs have been synthesized in liquid[41,42]. The ability of controlling the cyclodehydrogenations at selected molecular sites with an STM tip, even without a catalytic metal substrate or oxidants, provides an opportunity to synthesize freestanding GNRs and create novel intraribbon heterojunctions bottom-up.

## Discussion

We have established how the bottom-up synthesis of a graphene nanoribbon can be controlled by charge injections from an STM tip. From our experiments and first-principles calculations, it was found that the hole injections from an STM tip can trigger a cooperative domino-like cyclodehydrogenation even when the polymers are quasi-freestanding with suppressed substrate effect. The hole injections greatly reduce the energy barrier in the key step of the C–C bond formation. The H atoms migrate to the edge and dissociate into the vacuum as $H_2$ molecules. The cyclodehydrogenation process can be traced back to the classical Woodward–Hoffmann rules, showing that the formation of a C–C bond is symmetry allowed with hole injections but is instead symmetry forbidden with electron injections due to the phase mismatch of wavefunctions, corroborating the experimental observations. As the STM tip treatment can be performed at selective molecular sites without involving a catalytic effect from the metal substrate, the results point to a new way for bottom-up and controllable synthesis of freestanding GNRs and heterojunctions, which is critical for practical GNR-based nanodevices.

## Methods

**Sample preparation and STM measurements.** The Au(111) single crystal is cleaned by repeated cycles of argon ion bombardment and annealing to 740 K. DBBA molecules with a purity of 98.7% are used, which are degassed at 450 K overnight in a Knudsen cell (SVT Associates, INC.). Then, the molecules are evaporated at 485 K for 5 min from the cell with an effective coverage $\theta > 1$, while the Au substrate is held at 470 K. They dehalogenate upon adsorption. The sample is subsequently annealed at 470 and 670 K for 30 min, respectively, to induce colligation/polymerization (at 470 K) and cyclodehydrogenation/graphitization (at 670 K), resulting in polyanthrylene chains on 7-aGNRs. The STM characterizations are performed with a homemade variable temperature system at 105 K under ultrahigh vacuum conditions. A cleaned commercial PtIr tip is used. All STM images are acquired in a constant-current mode. The d$I$/d$V$ spectra are recorded using a lock-in amplifier with a sinusoidal modulation ($f = 1,000$ Hz, $V_{mod} = 20$ mV) by turning off the feedback loop-gain. The polarity of the applied voltage refers to the sample bias with respect to the tip.

**Calculation methods.** The *ab initio* calculations are performed with the Quantum Espresso code[43], using ultrasoft pseudopotentials[44] and Perdew–Burke–Ernzerhof (PBE) exchange correlation functional[45]. The PBE0 hybrid exchange correlation functional is used to correct the band gap[46]. The energy cutoff for the plane wave basis of Kohn–Sham wavefunctions is 24 Ry, and that for the charge density is 200 Ry. The structures are relaxed until forces on atoms reach a threshold of 0.026 eV Å$^{-1}$. The adsorption energies of the polymer and the GNR on the metal substrate are calculated by using a non-local van der Waals correction[47]. The charge density distributions are acquired as the square of the wavefunctions. The STM images are simulated based on Tersoff's method[48]. The energy barriers of the reaction are calculated using the NEB method[34]. The forces on images are relaxed until they reach a threshold of 0.1 eV Å$^{-1}$.

**Data availability.** The data that support the findings of this study, including the Supplementary Information, are available from the corresponding author A.-P.L. on request.

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

## Acknowledgements

This research was conducted at the Center for Nanophase Materials Sciences (CNMS), which is a DOE Office of Science User Facility. The electronic characterization was funded by ONR grants N00014-16-1-3213 and N00014-16-1-3153. The simulation work at NCSU was supported by DOE DE-FG02-98ER45685, with Z.X.'s work at CNMS being supported by the grant from Oak Ridge Associated Universities. The supercomputer time was provided by NSF grant OCI-1036215 at the National Center for Supercomputing Applications (NSF OCI-0725070 and ACI-1238993) and by DOE at the Oak Ridge Leadership Computing Facility and at the National Energy Research Scientific Computing Center. L.L. was supported by Eugene P. Wigner Fellowship at Oak Ridge National Laboratory.

## Author contributions

C.M. and Z.X. contributed equally to this work. A.-P.L. conceived the project and designed the experiments. J.B. and B.G.S. designed the theory tasks. C.M. and A.-P.L. performed characterizations; H.Z. and K.H. conducted molecule synthesis; Z.X., L.L., J.H. and W.L. performed the theoretical calculations. C.M. and A.-P.L. wrote the paper with contributions from all authors.

## Additional information

**Competing financial interests:** The authors declare no competing financial interests.

**Publisher's note**: 

