## [Peer Review File · Nature Communications]

Reviewers' comments:

Reviewer #1 (Remarks to the Author):

The manuscript entitled "Controllable conversion of quasi-freestanding polymer chains to graphene nanoribbons" describes locally controlled cyclodehydrogenation in the decoupled from metal surface polyanthrylene chain. To decouple the polymer from the metal surface, authors have performed a polycondensation of dibromobianthryl atop first-layer of GNRs that are in direct contact with the Au substrate. Authors have found that decoupling of the polymer greatly slow down the cyclodehydrogenation and a STM tip can be used to inject charges at the selected molecular sites to trigger desired cyclodehydrogenation leading to GNR formation. Additionally, a theoretical rationalization of the cyclodehydrogenation assisted by hole injections from STM tip is offered. The mechanism proposed is very similar to the Scholl reaction. Despite several uncertainty in the presentation of proposed mechanism, the cyclodehydrogenation via positively charged intermediated seems to be reasonable. (The way via cation radical should be calculated and discussed, please see remark 5). In general, the work provides interesting and very important data that allow deeper insight into the mechanism of cyclodehydrogenation and the mechanism of the Scholl reaction in particular (which is still not fully clarified). Moreover, this finding points a possible way toward controllable synthesis of GNR-based structures directly on insulating surfaces, which is of great interest. For this reason, I will recommend this manuscript for publication in "Nature Chemistry" after minor revision.

Minor remarks

1. Page 2, second line.

"...GNRs with variety of widths..."

Here the work on bottom up synthesis of the widest 15-GNRs (Carbon, 2014, 77, 1187-1190) has to be cited.

2. Page 8, line 2 (and line 5).

"...two phenyl groups on one side rotate to form C-C bond..."

These group are not a phenyl groups (C₆H₅) but benzyne groups (C₆H₄). Moreover these groups can't rotate. In this case the antryl unit is rotating around single C-C bond. Please correct.

3. Page 8, line 5.

"... migrated H atoms pushes the preexisting H to the back."

Such a description of the H migration process is not acceptable. Please change.

4. Page8, line 5.

"elimination of a H₂ molecules leads to the formation of a sp²-type C=C bond..."

An aromatic C-C bond is formed in this case . Please correct. (e.g. ...H₂ elimination leads to rearomatization of the system)

5. Page 8. Line 8.

Why barriers are calculated only for doubly charged systems? The single-hole injection would lead to the radical cation which seems to be an excellent candidate for the cyclization (at least according accepted mechanism of Scholl reaction). Moreover the localization of two charges in the conjugated systems such as polyanthrene seems to be highly unlikely.

6. The generalized mechanism for cyclodehydrogenation of bianthrylene molecule shown in figure 4c is a bit puzzling. (because it is generalized simultaneously for three different cases - cation, anion or neutral system). Should be shown for cation case only. Moreover, "similar" cyclization mechanism for polymer (negatively charged) given in SI looks different. In the first case the charge remain on the

PAH (elimination of H₂ molecule) in the second case two protons (H⁺) are leaving group. Please correct.

Reviewer #2 (Remarks to the Author):

I highly appreciated this manuscript discussing how graphene nanoribbons can be synthesized from quasi-free standing polymers by hole injections from a scanning tunneling microscope tip . Being a theoretician I mainly concentrated on the theoretical aspect of the work but credit should be given to the authors for the way they linked , also in the text , experiment and theory so that I could follow the overall strategy and argumentation .

Coming to the theoretical part the procedures followed are OK . The results are presented in a clear way and especially Figure 4 is a nice overview of the findings illustrating the big difference in activation barriers between hole and electron injection which is significantly lowered by hole injection as opposed to electron injection .This finding , in agreement with the experimental data , is nicely coupled to an orbital symmetry argument indicating that the two C atoms involved in bond formation show a negative overlap (out of phase) in the LUMO(LUCO) as compared to a positive overlap in the HOMO(HOCO) .This can indeed be traced back to the Woodward-Hoffmann type reasoning on symmetry allowed (hole injection) or forbidden (electron injection)reactions .Maybe I would not go so far saying that the Woodward Hoffmann rules are applied (bottom of p. 9 : "govern ") as the system considered is not one of the classical WH types .Maybe " traced back" is a better phrasing . Just one, also very minor detail,: the sentence on top of p.9, is not clear .The diagrams shown in Fig.4e are orbital plots and not density plots (they should be all positive in that case) .

Except from these minor points the paper can in my view be published as it stands, repeating however that I could not judge the experimental aspects .

Reply to reviewers' comments

Reviewer #1:

The manuscript entitled “Controllable conversion of quasi-freestanding polymer chains to graphene nanoribbons” describes locally controlled cyclodehydrogenation in the decoupled from metal surface polyanthrylene chain. To decouple the polymer from the metal surface, authors have performed a polycondensation of dibromobianthryl atop first-layer of GNRs that are in direct contact with the Au substrate. Authors have found that decoupling of the polymer greatly slow down the cyclodehydrogenation and a STM tip can be used to inject charges at the selected molecular sites to trigger desired cyclodehydrogenation leading to GNR formation. Additionally, a theoretical rationalization of the cyclodehydrogenation assisted by hole injections from STM tip is offered. The mechanism proposed is very similar to the Scholl reaction. Despite several uncertainty in the presentation of proposed mechanism, the cyclodehydrogenation via positively charged intermediated seems to be reasonable. (The way via cation radical should be calculated and discussed, please see remark 5). In general, the work provides interesting and very important data that allow deeper insight into the mechanism of cyclodehydrogenation and the mechanism of the Scholl reaction in particular (which is still not fully clarified). Moreover, this finding points a possible way toward controllable synthesis of GNR-based structures directly on insulating surfaces, which is of great interest. For this reason, I will recommend this manuscript for publication in “Nature Chemistry” after minor revision.

Reply: We highly appreciate the thorough review and insightful corrections by this reviewer.

Minor remarks

1. *Page 2, second line.*

“...GNRs with verity of widths...”

Here the work on bottom up synthesis of the widest 15-GNRs (Carbon, 2014, 77, 1187-1190) has to be cited.

Reply: We have now included this reference as Ref. [5], and corrected the typo from “verity” to “variety”.

2. *Page 8, line 2 (and line 5).*

“...two phenyl groups on one side rotate to form C-C bond...”

These groups are not a phenyl groups (C₆H₅) but benzyne groups (C₆H₄). Moreover these groups can't rotate. In this case the antryl unit is rotating around single C-C bond. Please correct.

Reply: We have corrected these mistakes on Page 8. The sentence at line 2 now reads as “**In the initial State 1, the neighboring anthrylene units first rotate around the single C–C bond, allowing two benzyne groups (C₆H₄) on the same side to form a single C–C bond giving int1.**” Accordingly, the sentence at line 5 is changed to “**Likewise, the benzyne groups on the other side repeat the process to form a graphitic lattice in State 3.**”

3. *Page 8, line 5.*

“... migrated H atoms pushes the preexisting H to the back.”

Such a description of the H migration process is not acceptable. Please change.

Reply: We have removed this sentence. The H migration process is already clear from the previous sentence.

4. *Page 8, line 5.*

“elimination of a H₂ molecules leads to the formation of a sp²-type C=C bond...”

An aromatic C-C bond is formed in this case. Please correct. (e.g. ...H2 elimination leads to rearomatization of the system)

Reply: The sentence has been corrected, and now it reads as “**Subsequently, the elimination of a H₂ molecule leads to the rearomatization of the system giving State 2.**”

5. Page 8. Line 8.

Why barriers are calculated only for doubly charged systems? The single-hole injection would lead to the radical cation which seems to be an excellent candidate for the cyclization (at least according accepted mechanism of Scholl reaction). Moreover the localization of two charges in the conjugated systems such as polyanthrylene seems to be highly unlikely.

Reply: We have now included the calculation results for single-charge injection systems, shown below and added in the SI as new Supplementary Fig. 11. We find that the barrier is reduced from 4.5 eV for the neutral case to 3.5 eV with a single-hole injection, where the reduction of 1.0 eV is much smaller than that of two-hole system (which is 4.5-2.8=1.7 eV, Fig. 4d). Moreover, there is a negligible energy barrier to prevent the **int1** state from going back to state **1**, unlike the two-hole injection case shown in Fig. 4d. In addition, the much reduced barrier for the two-hole injection system (2.8 eV) is comparable to the STM biases used to trigger the polymer-to-GNR reaction (–2.5 eV to –2.8 eV) (the SI text page 3). Thus, we believe the two-hole injection mechanism is more reasonable here. For the Scholl reaction mechanism, a second hole is needed to rearomatize the system too [Ref. 38, Grzybowski et al. Angew. Chem. Int. Ed. 52, 9900-9930 (2013)] (also see the new Supplementary Fig. 12, which is previous Supplementary Fig. 11). In the revision, we have mentioned these new results in the main text (page 8) and included the new figure in the Supplementary Information (Supplementary Fig. 11) along with a short description.

Supplementary Figure 11 | Energy diagrams of cyclodehydrogenation in vacuum for neutral (black), single-electron (green) and single-hole (red) assisted bianthrylenes, respectively, with the same reaction path as shown in the main text (Fig. 4d). With single-hole injection, the energy barrier can be reduced by about 1.0 eV from 4.5 eV (neutral) to 3.5 eV, while with single-electron injection, the barrier does not decrease.

As for the charge distributions in the polymer, our calculations show that the ground state would have the two charges delocalized in the polyanthrylene chain, as a result of the electrostatic repulsions between them. However, instantaneous localization of multiple charges can still happen as charges are continuously injected locally at one spot of the chain right under the STM tip during the period of pulse treatments.

6. The generalized mechanism for cyclodehydrogenation of bianthrylene molecule shown in figure 4c is a bit puzzling. (because it is generalized simultaneously for three different cases - cation, anion or neutral system). Should be shown for cation case only. Moreover, “similar” cyclization mechanism for polymer (negatively charged) given in SI looks different. In the first

case the charge remain on the PAH (elimination of H₂ molecule) in the second case two protons (H⁺) are leaving group. Please correct.

Reply: We show the reaction path for a generalized system without showing explicitly the charge states of the injected carriers in Fig. 4c, and then compare the energy diagrams of three different charge cases in Fig. 4d. If shown for the cation case only, Fig. 4c would look like the following.

Once holes are injected into State **1**, a C–C bond forms between the two benzyne groups at the top giving **int1**. For this intermediate, the two localized charges can be each represented as 4e/5c meaning four pi-electrons on five carbon centers. [1,3]-sigmatropic H migration gives **int2**, where the two localized charges can be represented as 3e/4c on the left and 5e/6c on the right. After H₂ elimination, the two holes are delocalized on the entire State **2**. The calculations in Fig. 4d were performed for the charged states. However, the two holes on State **2** may be dissipated to the metal substrate via the GNR lying underneath. With hole injections again, the two holes may be localized and then a C–C bond may form between the two benzyne groups at the bottom giving **int3**, where the two localized charges can be each represented as 4e/5c. Repeating the same [1,3]-sigmatropic H migration gives **int4** with 5e/6c on the left and 3e/4c on the right. Repeating the same H₂ elimination process gives State **3**, with the two holes delocalized on the entire molecule. The calculations in Fig. 4d were performed for the charged states. However, the two holes on State **3** may be dissipated to the metal substrate via the GNR lying underneath, giving neutral molecule, or neutral GNR in the case of extended system.

As can be seen above, if the charge state of $q=+2$ is considered in Fig. 4c, the figure itself and the discussions would become very complicated. For brevity, we choose to keep the original figure to represent a generalized mechanism for cyclodehydrogenation, with only minor changes on the locations of double bonds (cf. Fig. 4d in the original manuscript).

To clarify in the difference about the H elimination in the main text (Fig. 4c) and the SI (new Supplementary Fig. 12), we have added the following discussion in the Supplementary Information, page 3). It reads as “**The elimination of H in the form of either H₂ or 2H⁺ is dependent on the reaction environment, the gas phase in the UHV of the present study or the solution phase in the Scholl reaction. In the gas phase, our calculations for the bianthrylene case with $q=+2$ indicate that the elimination of H₂ is energetically more favorable than the elimination of 2H⁺. This is because charges prefer to be delocalized onto a greater domain of bianthrylene instead of being localized on two H⁺. However, for the Scholl reaction in a solution phase, the localized charges on H⁺ can be significantly stabilized by solvation energy. Thus we show H₂ elimination in Fig. 4c, and 2H⁺ elimination in Supplementary Fig. 12.**”

Reviewer #2:

I highly appreciated this manuscript discussing how graphene nanoribbons can be synthesized from quasi-free standing polymers by hole injections from a scanning tunneling microscope tip . Being a theoretician I mainly concentrated on the theoretical aspect of the work but credit should

be given to the authors for the way they linked , also in the text , experiment and theory so that I could follow the overall strategy and argumentation .

Coming to the theoretical part the procedures followed are OK. The results are presented in a clear way and especially Figure 4 is a nice overview of the findings illustrating the big difference in activation barriers between hole and electron injection which is significantly lowered by hole injection as opposed to electron injection. This finding , in agreement with the experimental data , is nicely coupled to an orbital symmetry argument indicating that the two C atoms involved in bond formation show a negative overlap (out of phase) in the LUMO(LUCO) as compared to a positive overlap in the HOMO(HOCO). This can indeed be traced back to the Woodward-Hoffmann type reasoning on symmetry allowed (hole injection) or forbidden (electron injection) reactions. Maybe I would not go so far saying that the Woodward Hoffmann rules are applied (bottom of p. 9: "govern ") as the system considered is not one of the classical WH types. Maybe "traced back" is a better phrasing.

Reply: We appreciate the reviewer's positive comments and recommendations. Following the reviewer's suggestion, we have revised the sentence mentioned by the reviewer from: "The Woodward-Hoffmann rules for orbital symmetry govern that the formation of a C–C bond is symmetry allowed with hole injections but is instead ..." to "**The cyclodehydrogenation process can be traced back to the classical Woodward-Hoffmann rules, showing that the formation of a C–C bond is symmetry allowed with hole injections but is instead ...**".

Just one, also very minor detail,: the sentence on top of p.9, is not clear .The diagrams shown in Fig.4e are orbital plots and not density plots (they should be all positive in that case) .

Reply: Actually, during our calculations using Quantum Espresso, the signs from the orbitals are passed to the charge density plots to give them different colors. To avoid confusion, we have followed the reviewer's suggestion and revised this sentence from: "Figure 4e shows the simulated charge density distribution of LUCO_p and HOCO_p in a polymer where the different signs passed from the orbital wavefunctions to the charge density plots are represented by blue and red colors, respectively." to: "**Figure 4e shows the simulated charge density distribution of LUCO_p and HOCO_p in a polymer, where the different signs as represented by blue and red colors are adopted from the orbital wavefunctions.**"

Except from these minor points the paper can in my view be published as it stands, repeating however that I could not judge the experimental aspects.

REVIEWERS' COMMENTS:

Reviewer #1 (Remarks to the Author):

All minor remarks have been satisfactorily addressed.